# Surface Structure Engineering of PdAg Alloys with Boosted CO_2_ Electrochemical Reduction Performance

**DOI:** 10.3390/nano12213860

**Published:** 2022-11-01

**Authors:** Xianghua Yang, Shiqing Wu, Qian Zhang, Songbai Qiu, Yuan Wang, Junjun Tan, Liang Ma, Tiejun Wang, Yongde Xia

**Affiliations:** 1Guangzhou Key Lab of Clean Transport Energy and Chemistry, School of Chemical Engineering and Light Industry, Guangdong University of Technology, Guangzhou 510006, China; 2School of Materials Science and Engineering, Sun Yat-Sen University, Guangzhou 510275, China; 3Department of Chemistry and Bioscience, University of Aalborg, Fredrik Bajers Vej 7H, 9220 Aalborg, Denmark; 4Faculty of Environment, Science and Economy, University of Exeter, Exeter EX4 4QF, UK

**Keywords:** electrocatalysts, bimetallic alloy, carbon dioxide reduction

## Abstract

Converting carbon dioxide into high-value-added formic acid as a basic raw material for the chemical industry via an electrochemical process under ambient conditions not only alleviates greenhouse gas effects but also contributes to effective carbon cycles. Unfortunately, the most commonly used Pd-based catalysts can be easily poisoned by the in situ formed minor byproduct CO during the carbon dioxide reduction reaction (CRR) process. Herein, we report a facile method to synthesize highly uniformed PdAg alloys with tunable morphologies and electrocatalytic performance via a simple liquid synthesis approach. By tuning the molar ratio of the Ag^+^ and Pd^2+^ precursors, the morphologies, composition, and electrocatalytic activities of the obtained materials were well-regulated, which was characterized by TEM, XPS, XRD, as well as electrocatalytic measurements. The CRR results showed that the as-obtained Pd_3_Ag exhibited the highest performance among the five samples, with a faradic efficient (FE) of 96% for formic acid at −0.2 V (vs. reference hydrogen electrode (RHE)) and superior stability without current density decrease. The enhanced ability to adsorb and activate CO_2_ molecules, higher resistance to CO, and a faster electronic transfer speed resulting from the alloyed PdAg nanostructure worked together to make great contributions to the improvement of the CRR performance. These findings may provide a new feasible route toward the rational design and synthesis of alloy catalysts with high stability and selectivity for clean energy storage and conversion in the future.

## 1. Introduction

Fossil fuel is an indispensable energy resource in modern society. However, the excessive use of fossil fuels has catastrophically resulted in the emission of a huge amount of carbon dioxide (CO_2_), which is one of the main greenhouse gases that lead to environmental pollution, climate change, and global warming [1,2]. Therefore, converting carbon dioxide into high-value-added products can effectively cope with the aforementioned problems, alleviating the environmental and climate crisis and benefiting the carbon cycle [3,4]. Currently, the reported CO_2_ conversion technologies include thermocatalytic [5], photocatalytic [6,7], electrocatalytic [8], and photoelectrocatalytic approaches [9,10]. In particular, electrochemical reduction of CO_2_ is considered the most promising way due to the tuneable reaction products and ambient working conditions [11,12]. Among the reported various products of CO_2_ reduction reaction (CRR), including carbon monoxide [13], formic acid [14], methane [15], ethanol [16], and ethylene [17], formic acid has attracted tremendous attention because it can be utilized as intermediate chemicals for industrial products [18], such as gloves, medical [19], and as hydrogen storage materials [20].

The most widely used electrocatalysts that can convert carbon dioxide into formic acid are Pb, Cd, Hg, In, Sn, Bi, and relevant compounds based on these metal ions [21,22,23,24]. Unfortunately, Pb, Cd, Hg, and In are poisonous to the environment, while Sn and Bi need to be carried out at a large overpotential of more than 300 mV [25]. As for Pd, previous research has found that Pd can convert CO_2_ into formic acid, but the in situ formed CO as the main byproduct during the reaction process is inevitably adsorbed on the catalytic active site of the Pd surface, which hinders the continuous reactions to further produce multi-carbon products [26,27]. Nevertheless, recent research has proved that alloying Pd with other metals can effectively upgrade its CRR activities and CO tolerance [28,29,30,31,32]. However, due to the symmetrical linear structure of the CO_2_ molecule and the highly uniform density of the electronic state of the whole molecule, CO_2_ is hardly adsorbed on the active sites of the catalysts with a bent configuration. Therefore, the capability of activating the chemical-adsorbed CO_2_ molecule into a curved structure on the active site for further reaction with e^−^ and H^+^ to form various products is essentially important [33]. A proper adsorption strength of the *CO intermediate produced during the CRR is crucial to the resulting products. Weak bound *CO intermediate to the metal surface may lead to fast *CO desorbed from the catalyst surface as the dominant product, while strong strength of the *CO intermediate may prevent further reduction of CO_2_, and consequently, hydrogen evolution reaction (HER) almost exclusively occurs. An optimized contact condition is also a key parameter to promote the CRR activities, which provides a fast electronic transfer rate and decreases the required energy for CRR via lessening the resistance among the liquid, the gas, and the solid interphase [34]. Hence, the rational design and synthesis of a Pd-based alloy electrocatalyst with superior CRR performance via the optimized contact angle and regulated surface structures with the desired proper adsorption strength of the *CO intermediate are theoretically and practically feasible.

Herein, we successfully developed a novel PdAg alloy with a two-dimensional morphology via a facile low-cost liquid synthesis approach using behentrimonium chloride (C_25_H_54_ClN) as a capping agent at ambient temperature and pressure. By tuning the composition of the precursors during the synthesis, the morphologies and chemical compositions of the catalysts were well-regulated. The CRR test confirmed that due to the optimized morphologies and chemical compositions, the Pd_3_Ag nanocrystals exhibited a higher electrochemical surface area, a faster electronic transfer speed, an enhanced CO resistance, and a lower charge-transfer resistance, which in all contributed to the increment of the CRR performance in varied aspects. This research demonstrated a simple and basic strategy to prepare and effectively regulate the CRR performance of the Pd-based nanostructures under mild conditions with the potential of mass production but less energy consumption. This research also provides a new economically available approach for the rational design and synthesis of novel catalytic materials with enhanced performance in clean and sustainable energy conversion, storage, and applications, which enables researchers to further expand the areas of their utilization in the field of electrocatalysis.

## 2. Materials and Methods

In a typical synthesis of Pd_3_Ag alloys, 0.335 g of PdCl_2_ (Sigma, 98%, St. Louis, MO, USA) and 20 mL of 0.2 mol/L HCl (Sigma, 37%) were mixed together and diluted to 200 mL to form a homogeneous chloropalladic acid solution with ultrapure water. Then, 0.5 mL of the as-obtained chloropalladic acid solution and 0.08 g C_25_H_54_ClN (98%) were all added into 10 mL ultrapure water under magnetic stirring for 30 min, followed by the injection of 0.167 mL 0.01 mol/L AgNO_3_ (Sigma, 99.9%) under room temperature and kept for 45 min; finally, 1 mL 0.3 mol/L L-ascorbic acid (C_6_H_8_O_6_, Sigma, >99.0%) was dropped into the above solution and maintained for 1 h. After the reaction, the product was collected through centrifugation at 8500 rpm for 10 min and washed with water three times and ethanol three times to remove most of the other species. Lastly, the product was dispersed in water for further characterization. The Pd_2_Ag, PdAg, and Pd syntheses followed similar procedures except for the different amounts of AgNO_3_. For the Pd_2_Ag synthesis, the added amount of AgNO_3_ was 0.33 mL and for PdAg was 0.5 mL. When no AgNO_3_ was added, the product was pure Pd crystals.

The as-obtained nanocrystal suspension was dispersed into ethanol under ultrasonic conditions. Then, the suspension was dropped on Mo grids coated with carbon film and dried under ambient conditions. Transmission electron microscopy (TEM) image was carried out on a HITACHI HT7700 operated at an acceleration of 100 kV. High-resolution transmission electron microscopy (HRTEM), scanning transmission electron microscopy (STEM) images, and energy-dispersive X-ray spectra (EDS) were all taken on an FEI, Thermo Talos F200S field-emission high-resolution transmission electron microscope operated at 200 kV.

Powder X-ray diffraction (PXRD) patterns were recorded on a Bruker D8 ADVANCE X-ray diffractometer with Cu-Kα radiation (λ = 1.54178 Å). X-ray photoelectron spectra (XPS) were collected on an ESCALab 250 X-ray photoelectron spectrometer, using nonmonochromatized Al-Kα X-ray as the excitation source. The concentration and ratio of palladium and silver nanoparticles were measured with a Thermo Scientific Plasma Quad 3 inductively coupled plasma mass spectrometry (ICP-MS) after dissolving them with a mixture of HCl and HNO_3_ (3:1, volume ratio).

The carbon dioxide reduction reaction (CRR) of the as-obtained samples was carried out in a liquid 0.1 mol/L KHCO_3_ solution (Sigma, 99.7%) system using a CHI760e electrochemical workstation (CH Instruments, Inc., Shanghai, China) equipped with an H-type electrochemical cell separated by a Nafion 211 membrane between the cathode and the anode at room temperature. An Ag/AgCl electrode (saturated with 3.5 mol/L KCl) and a Pt foil were used as the reference electrode and counter electrode, respectively. The glassy carbon electrode (GCE) with a diameter of 5 mm (working electrode area 0.196 cm^2^) was used as the working electrode (WE). To prepare the WE, 4 mg of catalyst and 20 μL of Nafion solution (5 wt%) were dispersed in 980 μL of an ethanol–water (3:1 *v/v*%) solution under ultrasonic conditions in a bath for 20 min to obtain a homogeneous ink. The ink was then dropped on the glassy carbon electrode and dried under ambient conditions with a loading amount of 1 mg/cm^2^. Prior to the CRR test, linear sweep voltammetry (LSV) was first carried out in the KHCO_3_ solution (0.1 mol/L) under a highly purified Argon atmosphere. For the CRR test, the LSV was recorded in the KHCO_3_ solution bubbled with a highly purified CO_2_ atmosphere at a constant rate of 20 mL/minute. The potential values of the Ag/AgCl reference electrode were calibrated with respect to RHE in all measurements using the following equation, E_vs_(RHE) = E_vs(Ag/AgCl)_ + 0.198 V + 0.059 pH. The quantification of gaseous CO_2_ reduction products was conducted using a gas chromatograph (GC, Agilent, 7890B) equipped with a thermal conductivity detector (TCD) and a flame ionization detector (FID). Liquid products were quantified using a Bruker AVIII 400 MHz ^1^HNMR (nuclear magnetic resonance spectroscopy). Typically, 500 μL of electrolyte was sampled at the conclusion of the electrolysis and was mixed with 100 μL D_2_O (Sigma, 99.9%), and 200 μL (m/m) DMSO (≥99.9%, Alfa Aesar, Haverhill, MA, USA) was added as the internal standard. The one-dimensional 1H spectrum was measured with water suppression using a pre-saturation method. The Faradic efficiency (*FE*) for formic acid (HCOOH) and CO were calculated via the same following equation:$$FE=\frac{2FVC}{Q}\times 100$$
where *F* is the faradic constant 96,485 C mol^−1^; *V* is the volume of the electrolyte obtained from the cathode; *C* is the concentration of the formic acid detected from NMR; *Q* is the total amount of the transferred charge.

*ECSA* was estimated from the CV curves in 1 mol/L KOH using the following equation:$$ECSA=\frac{Q_{PdO}}{0.405\;\mathrm{mC}\;m^{-2}\times {\;m}_{Pd}}$$
where Q*_PdO_* is the integral area of *PdO*, 0.405 is the charge required for *PdO* reduction, and *m_Pd_* is the *Pd* mass on the working electrode.

## 3. Results and Discussion

The TEM images of the Pd_3_Ag nanocrystals with varied magnifications in Figure 1a,b exhibit highly uniform two-dimensional nanostructures, with a diameter of nearly 50 nm. The high-resolution TEM image of Figure 1c shows the visible lattice fringes with a d-spacing of 0.221 nm, which matches well with the (111) plane of the Pd_3_Ag alloys [35,36]. Appendix A shows the d-spacing of 0.138, 0.198, and 0.112 nm indexed to the high-index- facets of (220), (200), and (222) of the Pd_3_Ag alloys in the corners, respectively; the selective area electronic diffraction (SAED) of the Pd_3_Ag in Appendix A also confirms the existence of the high-index-facets. The powder X-ray diffraction (PXRD) pattern of the as-synthesized Pd_3_Ag in Figure 1d shows five distinct diffraction peaks centered at 39.5°, 46.06°, 67.17°, 81.08°, and 85.58°, which closely corresponds to the face-centered cubic (fcc) Pd (JCPDS No.46-1043) but with an obvious shift of the diffract peaks position to low 2θ range. No other peaks ascribed to metallic Ag or Pd could be detected. Moreover, to investigate the effect of Ag content on the morphologies of the formed alloys, a controllable Ag level was introduced during the synthesis of alloy samples. As can be seen in Appendix A, when the molar ratio of Pd to Ag in the reaction system was 2:1, the morphologies of the Pd_2_Ag alloys evolved into a less branched structure (Appendix A). However, after further increasing the AgNO_3_ content to achieve the molar ratio of Pd:Ag to 1:1, the obtained PdAg alloy exhibited an irregular structure, with only 2~3 branches (Appendix A). When no AgNO_3_ was involved, and only the palladium precursor was added, the resulting Pd nanocrystals exhibited irregular pre-nanosheet morphologies, as shown in Appendix A. All these data firmly proved that the ratio of Pd to Ag in the precursor solution is the key parameter to tune the morphologies of the resulting nanocrystals. The XRD patterns of the nanocrystals obtained with a varied amount of AgNO_3_ presented in Appendix A also display the diffraction peaks located between the face-centered cubic pure Pd (JCPDS No.46-1043) and the face-centered cubic pure Ag (JCPDS No.04-0783), and the diffraction peaks of the alloy shifted to low 2θ and close to the diffraction peaks of fcc Ag phase, revealing the formation of the face-centered cubic (fcc) structure of PdAg alloys [35,37]. The high-angle annular dark-field scanning transmission electron microscopy (HAADF-STEM) and energy-dispersive X-ray spectroscopy (EDS) elemental mapping images (Figure 1e–h) proved that the Pd and Ag atoms were homogeneously distributed throughout the whole nanocrystals, further confirming the successful synthesis of PdAg alloys. In addition, the molar ratio of Pd:Ag was determined to be nearly 3:1 through inductively coupled plasma–mass spectrometry (ICP-MS), close to the calculated Pd and Ag precursor content in the synthesis protocol.

The atomic composition and chemical states of the PdAg alloys were all characterized using X-ray photoelectron spectroscopy (XPS). Figure 2a presents the full XPS spectrum of the representative alloy sample, which indicates the existence of Ag and Pd in the sample. The detailed XPS spectrum of Ag 3d in Figure 2b shows two peaks located at 367.3 eV and 373.3 eV that belong to Ag 3d_5/2_ and 3d_3/2_, respectively, which is about 0.3 eV positive shift, compared with the pristine metallic Ag. Moreover, no peaks were observed for Ag^+^, implying that only Ag^0^ existed in the nanocrystals. As shown in Figure 2b, the two peaks located at 334.7 eV and 340.05 eV were attributed to Pd 3d_5/2_ and Pd 3d_3/2_, which were about 0.3 eV positive shifts, compared with the pure Pd. Moreover, no peaks attributed to Pd^2+^ could be detected within the detection limits, suggesting that only the metallic state of Pd was present. All these results confirmed that the as-obtained product was a PdAg alloy, and both Pd and Ag were present in a bimetallic state in the sample, which is well in accordance with the XRD results of PdAg alloys [37,38].

Cyclic voltammetry (CV) studies were carried out to investigate the CRR performance of the as-obtained PdAg samples in 0.1 M KHCO_3_ solution saturated with Ar or CO_2_ in a potential range from −0.4 V to 0.15 V. As shown in Figure 3a, the current density of PdAg in the inert Ar atmosphere was much lower than that in the CO_2_ atmosphere because only hydrogen evolution reaction (HER) occurred under an Ar atmosphere. Moreover, the presence of CO_2_ was lower than that in the environment. The operation potential in CO_2_ was more negative with an increased current density from −0.22 mA/cm^2^ at 0 V to −6.69 mA/cm^2^ at −0.38 V, indicating that a CO_2_ reduction reaction occurred in this liquid system. The gas and liquid products for CRR were determined using GC and ^1^HNMR, respectively, after the CRR was carried out at a constant potential for 1 h (Appendix A). It is vital to note that only a trace amount of H_2_ and CH_4_ could be detected, and formic acid was the domain product in CRR (Appendix A). As can be seen from Figure 3b, the Pd_3_Ag electrocatalyst exhibited an initial faradic efficiency (FE) of 70% for formic acid at 0 V, and a maximum FE of 96% for formic acid at −0.2 V, which can be ascribed to the enhanced charge transfer speed resulting from the more negative potential. When further increasing the negative potential to −0.4 V, a mere FE of 60% for formic acid was obtained, an indication that −0.2 V was the optimized working condition.

A long-term durability test was carried out using the chronoamperometry method at −0.2 V for 7000 s. As can be seen from Appendix A, the current density of Pd_3_Ag nearly remained the same, which indicates that the morphologies, active site, and electronic transfer speed were maintained and remained unchanged. The HADDF-STEM (see Appendix A), EDX spectra, and EDS mapping images of the Pd_3_Ag collected after the long-term durability test show that the Pd and Ag atoms were homogeneously distributed throughout the whole nanocrystals, further confirming that the morphology and composition of PdAg bimetallic remained intact, suggesting the superb stability of Pd_3_Ag within a wide range of potential. For comparison, the selectivity of formic acid for Pd, Ag, and PdAg alloys with varied compositions was also investigated in the potential range of 0~−0.4 V, and the results are presented in Figure 3c. Pure Ag could not convert CO_2_ into formic acid and other chemical products at all, and only HER reaction occurred in this case. Pure Pd displayed only 30% selectivity for formic acid at a lower potential, and the highest selectivity of 85% could be reached at a potential of −0.2 V. However, the PdAg alloys exhibited an enhanced selectivity of 63% toward formic acid at 0 V, and the selectivity increased with the increase in the content of Pd in the alloys. The Pd_3_Ag sample demonstrated the highest selectivity above 96% in a potential range from −0.1 V to −0.3 V, which outperformed many reported CRR electrocatalysts (see Appendix A).

This remarkable enhancement might be due to the following three reasons: (i) Alloying the Pd with Ag can form two-dimensional nanosheet morphology, optimize the interface contact conditions between the catalysts and the electrode in a liquid system, and possibly decrease the resistance and accelerate the electronic transfer speed; (ii) the Ag atoms in the PdAg alloys can effectively alleviate CO adsorption on the PdAg surface and provide more catalytic active sites; (iii) the formation of PdAg alloys can effectively regulate the density of surface electronic structures to form both negative-dominate and positive-dominate domains, which can benefit the adsorption and activation of CO_2_ molecules from the line structures to a bent structure. The newly developed effective catalysts through the alloying of two metals in this work can remarkably increase the CRR efficiency and selection of targeted products.

The electrochemical active surface areas (ECSAs) for the four samples were all determined using the standard procedures from the integrated charge associated with the reduction of PdO to Pd. As shown in Figure 4a, the ECSAs for Pd_3_Ag, Pd_2_Ag, PdAg, and Pd were 57.5, 40.8, 35.4, and 11.1 m^2^ g^−1^, respectively, which indicates that alloying the Pd with Ag is an efficient approach to promote the CRR active sites. The enhanced ECSA for Pd_3_Ag may be ascribed to the high-index facets on the corners, which has been reported in previous research [39]. Tafel slope is a key inherent parameter to elucidate the rate-determined step of the CRR. As depicted in Figure 4b, the Tafel slope of Pd_3_Ag was 133 mV/decade (mV/dec), close to 143 mV/decade, indicating the formation of the CO_2_^•^-key intermediate is the rate-determining step for CO_2_ converting to formic acid [40]. However, the Tafel slopes of Pd_2_Ag, PdAg, and Pd were 168, 190, and 217 mV/dec, respectively, slightly larger than that of Pd_3_Ag, indicating that they have a similar rate-determining step with slower reaction rates.

The electrochemical impedance spectra (EIS) in Figure 4c show that the Pd_3_Ag electrocatalyst exhibited the smallest semicircle among the four samples in the Nyquist plot. The charge transfer resistance (Rct) values obtained by the semicircle in the high-frequency zone for Pd_3_Ag, Pd_2_Ag, PdAg, and Pd were 12.5, 33.7, 43.6, and 90.2 Ω, respectively, indicating that Pd_3_Ag had the lowest charge-transfer resistance at the catalyst/electrolyte interface and superior charge transport kinetics, which is consistent with the formerly observed result that Pd_3_Ag alloy exhibited the lowest Tafel slope and the fastest reaction rate due to its fast electronic and charge transfer in the CRR process.

## 4. Conclusions

In summary, we reported a facile room-temperature synthesis method to prepare highly uniform palladium–silver nanocrystals using C_25_H_54_ClN as the capping agent. By tuning the ratio of precursor Pd^2+^ and Ag^+^, the morphologies and electrocatalytic performance of the resulting PdAg alloys can be readily regulated. The palladium–silver alloy prepared with a molar ratio of 3:1 exhibited superior CRR activities with a FE of 69% for formic acid at a potential close to zero (−0.03 V), and the maximum FE up to 96% at −0.2 V without obvious current density decrease even after 7000 s reaction for CRR was achieved. In comparison, pure Ag showed almost no selectivity and FE for formic acid, and pure Pd exhibited only 69% selectivity for formic acid. These results demonstrate the superior activities of the Pd_3_Ag alloys toward CRR and the excellent selectivity for formic acid. Moreover, this type of alloy can be prepared via a very simple and low-cost inorganic synthetic route at room temperature in the aqueous phase. Thus, this cost-effect new synthesis method, combined with its promising performance in CRR, will extend our current knowledge in electrocatalysis and CO_2_ conversion and offer valuable materials and technical solutions for energy conversion and climate change.

## Figures and Tables

**Figure 1 nanomaterials-12-03860-f001:**
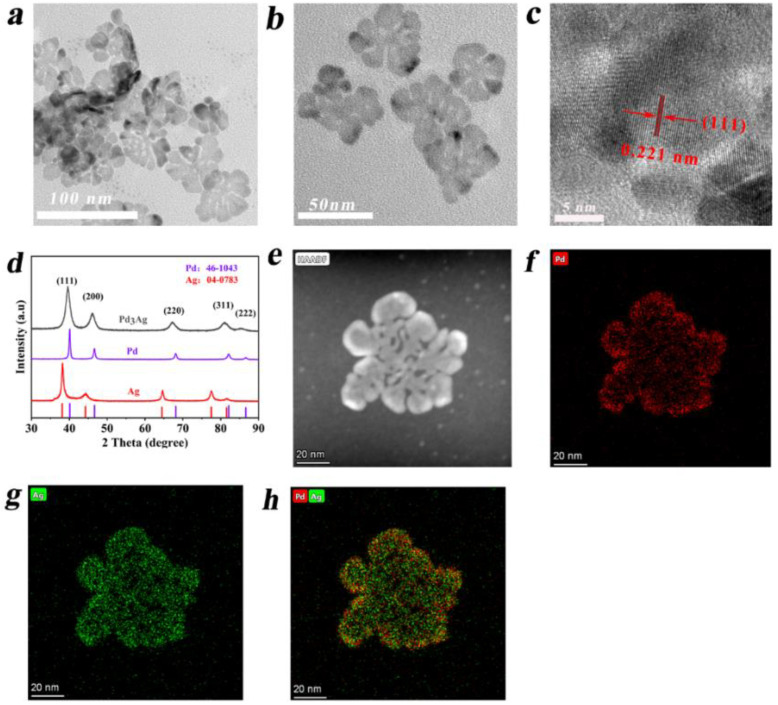
Structural characterizations of Pd_3_Ag nanosheets: (**a**,**b**) low-resolution TEM images; (**c**) high-resolution TEM images; (**d**) XRD patterns of Pd_3_Ag; (**e**) HADDF-STEM-EDS mapping of Pd and Ag in the selected area; (**f**) elemental mapping of Pd; (**g**) elemental mapping; (**h**) mixed pattern of Pd and Ag.

**Figure 2 nanomaterials-12-03860-f002:**
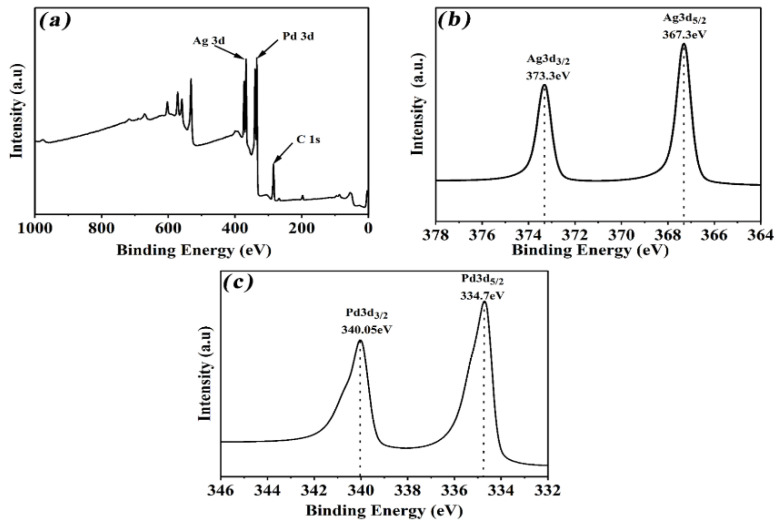
(**a**) Full XPS pattern of PdAg alloys; (**b**) high-resolution XPS spectra of Ag 3d; (**c**) high-resolution XPS spectra of Pd 3d.

**Figure 3 nanomaterials-12-03860-f003:**
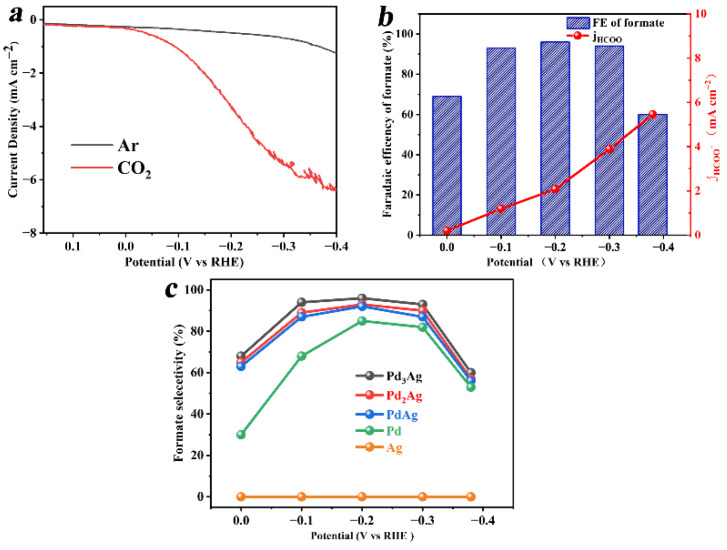
Electrochemical measurements of Pd_3_Ag: (**a**) polarization curves in Ar or CO_2_ saturated 0.1 M KHCO_3_; (**b**) potential-dependent FE for formic acid and partial current density; (**c**) selectivity for formic acid on Ag, Pd, PdAg, Pd_2_Ag, and Pd_3_Ag. The text continues here.

**Figure 4 nanomaterials-12-03860-f004:**
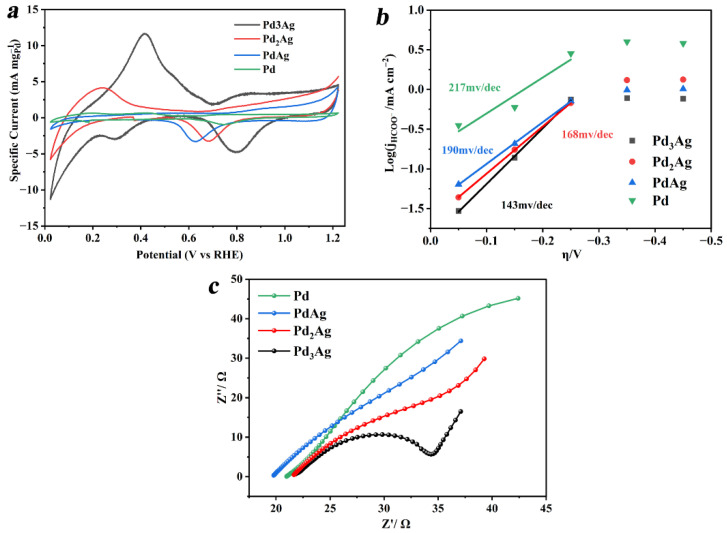
(**a**) ECSAs observed in N_2_-saturated 1 M KOH, (**b**) Tafel plot, and (**c**) electrochemical impedance spectra (EIS) in CO_2_-saturated 0.1 M KHCO_3_ of Pd_3_Ag, Pd_2_Ag, PdAg, and Pd alloy catalysts.

## Data Availability

All the data generated or analyzed in this manuscript are available in the article.

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
