# Peer review of "Surface Structure Engineering of PdAg Alloys with Boosted CO2 Electrochemical Reduction Performance"

_nanomaterials, 2022, doi:10.3390/nano12213860_

Round 1

Reviewer 1 Report

I have carefully read this paper entitled with “Surface Structure Engineeringed PdAg Alloy with Boosted CO2 Electrochemical Reduction Performance ". As a result, I have only a few minor points that the authors should address before it is accepted for publication. Please, publish subject to the following revisions:

1-     Rewrite the novelty statement at the end of introduction section.

2-     Authors should justify the importance of the current work of how this is different from earlier reports. So, it’s better add comparison table material and its performance to shows importance of manuscript.

3-     Why the Electrochemical impedance spectra (EIS) of the electrodes have two semicircles.

4-     Please add CV with different scan rate data for estimating the electrochemical active surface areas (ECSA)

Reviewer 2 Report

Within the article, the authors presented PdAg alloys with engineered surfaces in order to enhance their performance during CO2 electrochemical reduction reactions. The results shown in the manuscript are well presented and can be useful for further research within this area of research. Thus I recommend publishing the manuscript after minor corrections:

The quality of the STEM image and  EDX maps presented in SI Figure S8 is really poor, thus it’s not sufficient to confirm that the Pd3Ag alloy preserved previous structure and morphology after a long-term durability test. This experiment must be repeated. Moreover, a set of EDX maps must be accompanied by EDX spectra to confirm the presence of given elements.
